# New perspectives on university quality assessment: A Mamdani Fuzzy Inference System approach

**Cristina Carrasco-Garrido**[1], **Belen Maria Moreno-Cabezali**[1], **Antonio Martínez Raya**[2*]

**1** Department of Business Economics, Applied Economics II, and Fundamentals of Economic Analysis, Rey Juan Carlos University—Universidad Rey Juan Carlos (URJC), Madrid, España, **2** Department of Organizational Engineering, Business Administration and Statistics, Technical University of Madrid — Universidad Politécnica de Madrid (UPM), Madrid, España

* antoniomartinez@upm.es

## Abstract

Higher education has traditionally played the role of an overarching factor in economic growth and development. The implementation of the European Higher Education Area (EHEA) has already achieved improvements in many educational areas, but there remain, within the requirement to ensure academic excellence, cases where the quality criteria are not entirely harmonized. Genuine harmonization among the 48 countries that have so far been affiliated with the EHEA has been a key challenge for national educational assessment agencies and related bodies. This study aims to analyze the quality of the Spanish university system partially through a model based on the Mamdani Fuzzy Inference System (FIS) methodology. Numerous studies have been identified that evaluate university quality from the perspective of the student, but there are no studies that analyze the quality of public higher education institutions from the perspective of faculty employees. This research gap prompted an extensive literature review, considering fifteen main elements classified into five categories: internationalization; scientific production, occupational category, academic background, and professional experience. Researchers collected and curated data from a database of four Madrid-based public institutions. A Mamdani FIS, yielding a unique assessment in each case, was implemented using the MATLAB Fuzzy Logic Toolbox. Therefore, the results have been evaluated to determine which institution has led to better educational quality. The research approach leads to measuring the quality of public higher education institutions. First, thanks to the quality evaluation from the perspective of the workers and the professors who are part of the four public universities in Madrid. Second, we carried out this analysis under a methodology that has not been used before on that issue. Concerning its practical implications, this study can help policymakers design better practices to improve the careers of university professors and, as a result, the quality of higher education and the future employability of graduates.

**Data availability statement:** All relevant data are within the manuscript and attached files, namely Supporting Information and raw data.

**Funding:** The payment of APC was managed by Cristina Carrasco-Garrido and then settled fully by the Rey Juan Carlos University—Universidad Rey Juan Carlos (URJC) [ROR: https://ror.org/01v5cv687]. There was no additional external funding received for this study.

**Competing interests:** The authors have declared that no competing interests exist.

## Introduction

The university world is becoming increasingly competitive, and the number of public and private universities is growing. A case study conducted in the United Kingdom over 10 years [1] determined the quality of teaching as one of the predictors of the general satisfaction of university students. For this reason, there must be characteristics that allow one to assess the quality of universities.

In the European context, the concept of quality in universities has occupied a central place. Quality assessment in European Higher Education Institutions (HEI) became a key issue among academics and policymakers [2]. For several years, universities have assessed implicit academic reputation, since the exponential growth of the international higher education market has encouraged the need for rigorous comparisons with other institutions [3].

The newly committed quality policy, which had just arrived, had the clear objective of establishing quality assurance mechanisms. In 2003, the European Association for Quality Assurance in Higher Education (ENQA) was established. Since then, there has been a rapid proliferation of Quality Assurance Agencies (QAAs) that oversee quality assurance in HEIs [4]. In fact, many countries have developed a way to assess the quality of HEIs. Spain is a prime example of this phenomenon.

Quality assurance in higher education is a generic term open to many interpretations. Since quality is subjective, any definition of university quality should draw upon as many points of view as possible [2]. The result of all the efforts made by the literature has culminated in a wide variety of ranking systems. Despite the rapid proliferation of international ranking systems in recent years, criticisms of cross-border university comparisons have emerged [2]. Differences in the history, culture, educational traditions, and perspectives of universities led some authors to argue that comparing universities can be problematic because sources that differ from some universities to others affect quality [5,6].

The existing literature has not reached an agreement on the best methodology for the quality assessment of universities. In addition, most studies focus on the student's perspective, analyzing their satisfaction and perception [7–9]; leading to the lack of research focused on the characteristics of human capital. In this sense, this paper proposes a novel methodology for the quality assessment of universities. Using a Mamdani Fuzzy Inference System, the study presents a procedure based on the characteristics of the human capital of HEIs. The proposed metrics are based on five dimensions: internationalization, scientific production, occupational category, academic background, and professional experience.

The paper adheres to the following structure: Section 2 presents the conceptual framework; Section 3 introduces the data analysis and definition of variables; Section 4 focuses on the design and application of the model; Section 5 shows the results; and Section 6 concludes. Finally, in Section 7, the researchers discuss limitations and future research.

## Conceptual framework

### Background

**The emergence of evaluation agencies in the European context.** European Higher Education policies have had important transformations during the last two decades. In 1999, education ministries set the goal of creating a European Higher Education Area (EHEA), which has led the higher education systems of European countries to introduce important changes in their structure, practices, and cultures [4]. Its clear commitment to quality and the establishment of quality assurance mechanisms characterized this newly arrived policy [10].

The Bologna Declaration [11] became a turning point concerning program harmonization and quality regulation of European higher education. To achieve this goal, the European ministers established the objective of setting the EHEA by 2010. The creation of a common assessment framework, based on a European network for quality assurance, was an idea that was already beginning to sound. In 2003, the European Association for Quality Assurance in Higher Education (ENQA) was founded to promote cooperation in the field of quality assurance. This association oversaw the implementation of accreditation procedures. At the same time, the conviction began to spread that accreditation has been the clear seal of quality and that the entire quality assurance process in higher education institutions should be carried out by autonomous Quality Assurance Agencies (QAAs), formally separated from governments and universities [4].

This policy process that was being carried out resulted in the proliferation of QAAs in Europe. The quick implementation of the reforms related to the Bologna Declaration required a rapid harmonization of the quality assessment regulations and practices recently introduced. Accreditation of university degrees became progressively compulsory, leading QAAs to be placed at the head of this new governance system. Agencies became the ones in charge of introducing accreditation, the just-key ingredient [4].

**The law of evolution in the case of Spain.** The National Agency carries out the quality assurance of Spanish higher education institutions, both public and private, out for Quality Assessment and Accreditation, together with 11 regional agencies. The National Agency for Quality Assessment and Accreditation (ANECA) was established in 2002 because of the Spanish Universities Act [12], which required the creation of an independent external body for quality assurance in the Spanish higher education system.

Since its creation, ANECA has undergone substantial transformations, influenced by the progressive construction of the EHEA. The lack of specification of the approaches to be applied in the evaluation of higher education within the legislative framework determined 'the commitment of the ANECA governing bodies to organize its evaluation processes according to standards and guidelines for quality assurance in the European Higher Education Area (EHEA), according to the report of ANECA [13]. That same year, Spanish quality assurance experienced a crucial transformation: the transition from voluntary to compulsory. In 2007, a new system of qualifications was established, and the system of accreditation for university degrees became compulsory [14]. Thus, ANECA launched programs for the accreditation of all degrees in Spain within three years. Furthermore, a national system was introduced to accredit Spanish university educators [4].

To favor the assessment of academic staff, ANECA launched the Support Program for the Evaluation of Teaching Activity of University Faculty, commonly known as DOCENTIA. This program aimed to support universities in the design of their mechanisms to manage the quality of the teaching activity of university teaching staff and favor its development and recognition [15]. DOCENTIA takes the recommendations for quality assurance in higher education institutions, approved by the Conference of Ministers signatories of the Bologna Declaration in May 2015 [16]. It is presented as a system for the evaluation of teaching staff, which serves as a basis for universities and autonomous universities and regional agencies to design their evaluation models, combining information from different sources, organized into dimensions and criteria, to which weights are assigned to obtain the final evaluation [4].

After all this transformation and creation process of the new regulatory framework, the professional career in Spanish universities is structured into a total of five professional categories, which, in this study, will be divided between tenured

workers in the public sector and non-tenured workers to reduce the complexity of this new structure. For each position, this is a must but not sufficient to obtain the corresponding accreditation from the ANECA.

### Theoretical aspects

**The emergence of evaluation agencies in the European context.** Quality in higher education is a concept that has been difficult to define in the literature. Gibson [17] stated that 'quality is not easier to describe and discuss than to deliver in practice.' The wide range of functions universities must perform leads to a lack of consensus on the best way to define and measure university quality [18].

Quality assurance in higher education is a generic term open to many interpretations. Since quality is subjective, any definition of university quality should draw on as many different points of view as possible [2]. The result of all the efforts made by the literature has culminated in a wide variety of classification systems.

In the case of Spain, the authors of Repáraz et al. [19] proposed six indices that aim to reflect the quality of Spanish universities. These indices are based on 71 quality indicators. One of those indices is about human capital. Evaluation is continually being studied, as it is considered an important task since it leads the direct path to educational quality [20,21].

In the paper, a system for measuring the quality of some universities based in Madrid has been introduced according to certain characteristics of human capital. The proposed methodology could be applied to similar cases at hand, mainly in the Spanish context. Five dimensions form the related metric: internationalization, scientific production, occupational category, academic background, and professional experience.

Another study conducted in China, proposed increasing transnational cooperation in higher education as an enriching element to improve the quality control system in higher education [22].

Another dimension of quality in HEIs is scientific production [23] state that quality in HEIs can be indirectly measured through intellectual and scientific production. Scientific production has become a fundamental instrument for improving academic quality in HEIs worldwide [24]. Therefore, scientific production has traditionally been considered a fundamental pillar of higher education in all countries.

In Spain, which belongs to ANECA, there is a national body named CNEAI (Comisión Nacional Evaluadora de la Actividad Investigadora), which is responsible for evaluating the research activity of university research and teaching staff, to grant them a productivity incentive [13]. This evaluation is carried out over three, five, and six-year research periods with high-impact publications. The 3-year periods are called triennium, the 5-year periods are called quinquennium, and the 6-year periods are called sexennium. Therefore, these periods can be considered as an indicator of the scientific production of the teaching staff and will be used in this study to evaluate this dimension.

The third dimension is the occupational category. Using a qualitative methodology, Hortigüela Alcalá et al. [25] showed that university quality is closely related to the occupational category of academics. In the Spanish university system, occupational categories are divided into two groups: contracted staff (non-tenured workers) and civil servants (tenured workers). In this way, dividing university instructors will study this variable into these two categories. These two groups are, in turn, subdivided into four different figures (two within each group), as follows:

-Non-tenured workers: the university career in the Spanish system begins in this first group with the position of assistant professor, a so-called 'profesor ayudante doctor', whose position is habitually eligible to be promoted to the next level of an employment contract as associate professor, known as 'profesor contratado doctor'.

-Tenured workers: this second group includes the positions of associate professor 'profesor titular de universidad' and 'catedrático de universidad'.

Together with these three dimensions, the literature has also been considered an important variable in assessing the quality of higher education faculty of academic background. To study this variable, the sample will be divided into teaching

staff with a Ph.D. and those with another type of education (e.g., master, degree, etc.). Ph.D. is considered the period in which attitudes and values of academics are developed. This period, which normally takes around 3–4 years, is focused on developing independent researchers. Through this long process of supervised study, most academics are prepared for appointment to teaching and research roles. Probert [26] considers that even those who are scheduled because of their professional experience will find it difficult to make their way further up the academic hierarchy unless they embark on a Ph.D. In their study, Pehmer et al. [27] suggest that HEIs should only recruit faculty who meet three requirements. One of them is having an in-depth research competence, which is mostly acquired through the completion of a Ph.D. [28].

As introduced below, it constitutes the fifth and last dimension of the quality assessment system proposed in this paper: professional experience. In Spain, the university system is very strict with the full-time dedication of the teaching staff; however, there is a hybrid role that combines part-time work at the university with work outside the university, a mixed figure that was created to provide a purely professional vision of the different fields of study. Studies such as the one conducted by Quiroz Pacheco & Franco García [29] show that the knowledge transmitted to students by professors with extensive professional experience is more enriching and meaningful compared to professors with little or no work experience. When asked by students, they prefer teaching based on contextualized learning rather than simply lectures [25].

In recent years, Spanish universities have been immersed in profound legislative changes that modify some of the current positions, affecting the current and future contracts of a large part of the teaching staff and their performance [30]. Therefore, during the academic year 2023–2024, some of them even required modifying certain employment contracts of school staff. To ensure that these changes made during the transition period of the new law do not have an impact on this work, the data for the year 2022 have been taken as a reference. However, it is important to take these into account because they will be discussed throughout this work.

## Analysis of data and definition of linguistic variables

A total of four Spanish public universities have been the subjects of this study: Universidad Complutense de Madrid (UCM), Universidad Rey Juan Carlos (URJC), Universidad Autónoma de Madrid (UAM), and Universidad Carlos III (UC3M). Data for these universities have been extracted from UniversiDATA (more information about this database is available on the UniversiDATA website: https://www.universidata.es). Its purpose is to provide open data from the higher education industry in Spain for use and value by universities, the media, and society in general [31]. To understand the analysis carried out and extrapolate it to other countries, as shown in Table 1, it has been concluded that the positions regarding university research and teaching staff in Spain and the United States are similar.

Although not all categories exist in each university, to be consistent throughout the research and to guarantee the correct comparison between the four universities in this study, it has been decided to use only the common figures: profesor catedrático, profesor titular, profesor contratado doctor´, `profesor ayudante doctor´and `profesor asociado´. For the study, they are classified into tenure and non-tenure, as shown in Table 1. To analyze the quality of university teaching, five relevant variables have been identified in Table 2.

University professors' knowledge, skills, and abilities determine elements of their quality, and therefore of the universities of which they form part.

These are intangible elements, which are difficult to quantify. The ANECA is responsible for reviewing and assessing whether professors meet specific and previously determined merits, intending to accredit them to specific figures.

Therefore, we can affirm that if in this study the occupational category is analyzed and this is previously granted by ANECA, the knowledge, skills, and abilities of the professors of the universities evaluated here are implicitly included in the study.

The first variable is internationalization. In this paper, a differentiation has been made between national faculty and personnel with a nationality other than Spanish. The second variable is scientific production, which will be measured through the mechanism used by the state agency ANECA to measure scientific performance through the sexennium of research

**Table 1. Faculty staff equivalences between Spain and the U.S.**

| Spain (in Spanish) | United States of America (in US English) | Contract term |
|---|---|---|
| Profesor catedrático | Professor/Full professor | Tenure |
| Profesor titular | Associate Professor | |
| Profesor contratado doctor | | |
| Profesor colaborador | | |
| Profesor ayudante doctor | Assistant Professor | Non-Tenure |
| Profesor asociado | Professional Professor | |
| Profesor visitante | Visiting Professor | |
| Profesor emérito | Emeritus Professor | |

Source: own elaboration.

**Table 2. Linguistic variables and categories.**

| Variables | Factors |
|---|---|
| Internationalization | National |
| | Abroad |
| Scientific Production | Triennium |
| | Quinquennium |
| | Sexennium |
| Occupational category | Tenure |
| | Non-tenure |
| Academic background | Ph.D. |
| | Other training |
| Professional experience | Combines teaching with another job. |
| | Does not combine teaching with another job |

Source: own elaboration.

and, in its case, the Community of Madrid through the sexennium, quinquennium, and triennium. The sexennium evaluates the five best contributions made during six years [32]. The quinquennium evaluates the four most relevant contributions with the highest impact made for five years [33], and the triennium evaluates the three most relevant publications with the highest impact made for three years. On the one hand, academic background is a variable that has been divided between those who have a Ph.D. and those who do not. On the other hand, the professional category has been divided into tenure and non-tenure staff, as shown in Table 1. The fifth and last variable is that that establishes professional experience. This variable is relevant because it highlights the difference between faculty staff who are dedicated exclusively to teaching and other types of higher education faculty known as 'associate professors' in Spain. These are professionals who are experts in the subject they teach. As a result, students obtain this theoretical-professional duality, reflected in the teaching quality.

## Design and application of a quality assessment model for higher education

The study focuses on developing a model to evaluate the educational quality of four public institutions in Madrid from the perspective of academics. This technique can be highly effective in identifying areas for improvement in teaching approaches. Its application can enable universities to gather relevant information on teaching quality, thereby improving data-driven decision-making to improve student learning experiences. In addition, it can serve as a precedent for the

development of policies that promote teaching careers and improve the quality of higher education, ultimately improving the future employability of graduates.

Fig 1 illustrates the steps taken to design and implement the model. The model was developed using a fuzzy inference system (FIS) within the MATLAB Fuzzy Logic Toolbox. This tool facilitates the design, simulation and analysis of FIS, allowing the configuration of inputs, outputs, membership functions, and rules for both type-1 and type-2 FIS. The software provides a flexible and powerful platform for implementing algorithms based on fuzzy logic [34].

The FIS employed in this study is grounded in fuzzy set theory, which was developed by Lotfi A. Zadeh in the 1960s. Fuzzy set theory is a branch of mathematics that addresses intricate and ill-defined problems arising from incomplete or imprecise information. This approach is effective for uncertain or approximate reasoning, particularly when intuitive human thinking is involved [35].

In classical (crisp) set theory, any element x in a universal set X could belong or not belong to a subset A, denoted as A ⊆ X. A membership degree can be defined using a characteristic function. It assigns a binary value (either 1 or 0) to each element $x \in X$, where 1 denotes membership and 0 indicates non-membership [36]. In contrast, fuzzy logic uses fuzzy sets instead of crisp sets. A fuzzy set is a class of objects that have a continuum of degrees of membership. A membership function (characteristic) assigns a degree of membership to each object ranging from zero to one [37].

The proposed model estimates the level of university education (output variable) based on the final scores obtained for each of the five input variables: internationalization, scientific production, occupational category, academic background, and professional experience. The evaluation of the 15 factors is performed integrally, resulting in a unique final score for each university.

The following are the steps involved in the development of the FIS used as a model to evaluate the quality of higher education.

## Selection of the FIS type

The first step in creating the quality evaluation model is to decide on the type of FIS. An FIS is a fuzzy logic-based system that links inputs and outputs using logical operations and base rules [38]. The MATLAB Fuzzy Logic Toolbox allows users to choose between two types of FIS: Mamdani and Sugeno systems. The FIS used in the study is based on the Mamdani type due to its simplicity, ease of formalization, high interpretability, widespread acceptance, and suitability for human input [39]. The related advantages are summarized in Table 3.

## Selection of the linguistic variables

The second step involves defining the linguistic variables. A linguistic variable is one whose values are expressed as words or sentences rather than numbers. Although linguistic values are less accurate than numerical values, they better represent human cognitive processes. According to Rodríguez Rodríguez et al. [40], linguistic variables can be used to describe complex or unclear objects that are difficult to quantify numerically.

The model comprises five input variables: internationalization, scientific production, occupational category, academic background, and professional experience (see Fig 2). Calculations based on these input variables produce an output

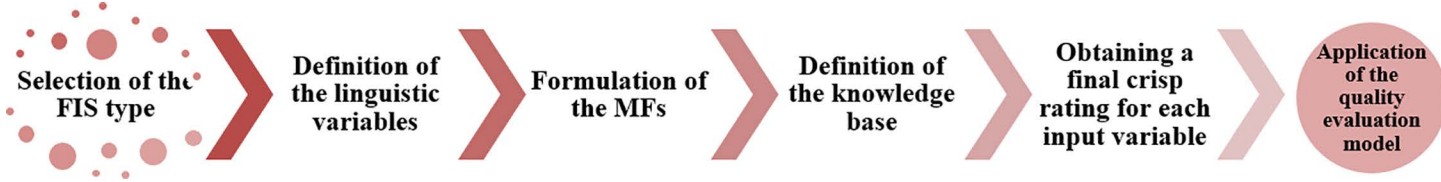

**Fig 1. Design phases of the quality evaluation model.** Source: own elaboration.

**Table 3. Advantages of each type of FIS.**

| FIS | Advantages |
| --- | --- |
| Mamdami | Intuitive |
| | Adaptable to human input. |
| | The more interpretable rule base |
| | Have widespread acceptance |
| Sugeno | Computationally efficient |
| | Work well with linear techniques (e.g., PID control) |
| | Work well with optimization and adaptive techniques. |
| | Guarantee output surface continuity |
| | Suitable for mathematical analysis |

Source: own elaboration based on information collected from [34].

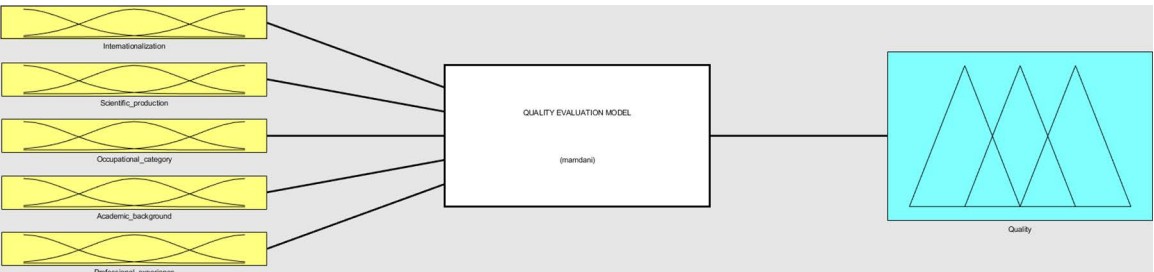

**Fig 2. Structure of the quality evaluation model.** Source: MATLAB Fuzzy Logic Toolbox Software (MATLAB 9.14 R2023a).

variable: the level of quality of higher education. As presented above (see Section 2), these variables were selected based on a thorough review of the literature.

## Formulation of the membership functions (MFs)

The third step involves formulating the appropriate MFs for fuzzy sets of linguistic variables. A membership function is a mathematical function that quantifies the inclusion of an element in a fuzzy set, using linguistic concepts rather than exact numerical values [41].

Various types of MFs exist, such as triangular, trapezoidal, Gaussian, etc. Although the MATLAB Fuzzy Logic Toolbox provides different classes of MFs, triangular functions were chosen for this model. Along with trapezoidal functions, triangular functions are favored due to their simplicity, robustness, ability to model reality, and computational efficiency [42].

The model consists of five input variables and one output variable. For each linguistic variable, three fuzzy sets ("Low", "Moderate" and "High") were established, represented by three linguistic terms, and defined using three triangular MFs. Fig 3 graphically depicts the MFs of the fuzzy sets for each linguistic variable, as well as their associated linguistic terms.

Equation 1 defines the triangular MF (see Fig 3) for each fuzzy set as follows:

$$\mu_A(x) = \begin{cases} 0, & \text{if } x \leq a \\ \frac{x-a}{b-a}, & \text{if } a < x \leq b \\ \frac{c-x}{c-b}, & \text{if } b < x \leq c \\ 0, & \text{if } x > c \end{cases}$$

(1)

For $a < b < c$.

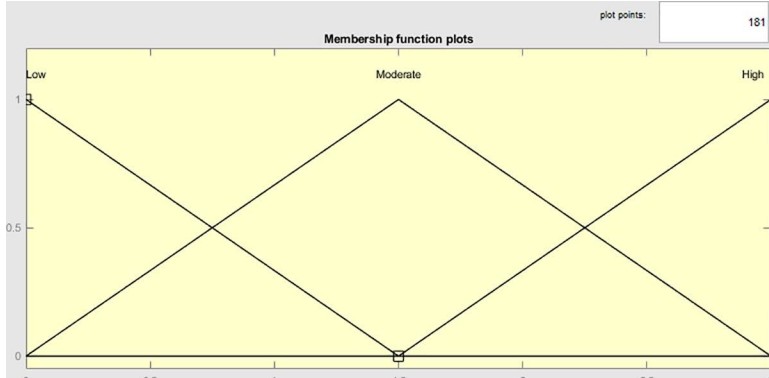

**Fig 3. Graphic representation of the MFs of each linguistic variable.** Source: MATLAB Fuzzy Logic Toolbox Software (MATLAB 9.14 R2023a).

Linguistic terms, fuzzy triangular numbers, crisp ratings on a three-value scale, and the general interpretation of the fuzzy sets for each linguistic variable are presented in Table 4.

## Definition of knowledge base

The fourth phase required the creation of a fuzzy rule base for the FIS. The fuzzy inference process operates on the basis of if-then rules. The established rules define the relationship between the input parameters and the output result [43]. These rules consist of two major components [34]:

- Antecedent: Also known as the premise, it represents the 'if' component of the rule and specifies the linguistic terms of the input variable.

- Consequent Often identified as the conclusion, it represents the 'then' component of the rule and specifies the linguistic terms of the output variable.

The Rule Editor in the MATLAB Fuzzy Logic Toolbox is used to incorporate these if-then rules. To define the antecedent of the rule, it is necessary to specify the connection operator. The connection parameter can have one of these two values [34]:

- AND: Connects the input linguistic terms using an AND operation.

- OR: Connects the input linguistic terms using an OR operation.

In this study, the FIS was designed using the AND connection parameter. A total of 243 rules were used to describe the direct causal relationship between the input variables (internationalization, scientific production, occupational category, academic background and professional experience) and the output variable (quality of higher education). Each rule was assigned equal importance. Fig 4 illustrates the configuration of the 243 if-then rules applied in the model for assessing the quality of higher education.

## Determining a crisp rating for each input variable

The fifth stage aimed to perform a precise assessment of each input variable using a three-value scale. According to the parameters outlined in Table 2, this involved performing a detailed proportional evaluation of the factors associated with each variable as follows.

**Table 4. Description of linguistic terms, crisp ratings, and fuzzy triangular numbers associated with fuzzy sets.**

| Variable | Crisp rating | Linguistic term | General Interpretation | Triangular fuzzy number |
|---|---|---|---|---|
| Internation-alization | 1 | Low | Internationalization is deemed low when fewer than one-third of the teaching staff are non-Spanish speakers. | (−1.5 0 1.5) |
| | 2 | Moder-ate | Internationalization is deemed moderate when between one-third and two-thirds of the teaching staff are non-Spanish speakers. | (0 1.5 3) |
| | 3 | High | Internationalization is deemed high when more than two-thirds of the teach-ing staff are non-Spanish speakers. | (1.5 3 4.5) |
| Scientific Production | 1 | Low | Scientific production is classified as low when the count of trienniums, quin-quenniums, and sexenniums is less than 4,250 in each category. | (−1.5 0 1.5) |
| | 2 | Moder-ate | Scientific production is classified as moderate when the count falls within the range of 4,250–8,500 in each category. | (0 1.5 3) |
| | 3 | High | Scientific production is classified as high when the count exceeds 8,500 in each category. | (1.5 3 4.5) |
| Occu-pational category | 1 | Low | The occupational category is classified as low when less than one-third of university lecturers hold tenured positions. | (−1.5 0 1.5) |
| | 2 | Moder-ate | The occupational category is classified as moderate when one-third to two-thirds of university lecturers with tenured positions. | (0 1.5 3) |
| | 3 | High | The occupational category is classified as high when more than two-thirds of university lecturers hold tenured positions. | (1.5 3 4.5) |
| Academic background | 1 | Low | Academic background is considered low when fewer than one-third of lectur-ers with Ph.D. | (−1.5 0 1.5) |
| | 2 | Moder-ate | Academic background is considered moderate when between one-third and two-thirds of lecturers with Ph.D. | (0 1.5 3) |
| | 3 | High | Academic background is considered high when more than two-thirds of lecturers with PhD. | (1.5 3 4.5) |
| Professional experience | 1 | Low | Professional experience is considered low when less than one-third of aca-demics combine teaching with another job outside the university. | (−1.5 0 1.5) |
| | 2 | Moder-ate | Professional experience is considered moderate when one third to two thirds of academics combine teaching with another job outside the university. | (0 1.5 3) |
| | 3 | High | Professional experience is considered high when more than two-thirds of academics combine teaching with another job outside of the university. | (1.5 3 4.5) |

Source: own elaboration

- Internationalization. This variable consists of two components: Spanish academics and those from outside Spain. Measure the proportion of foreign faculty members.

- Scientific production. This variable includes three components: triennium, quinquennium, and sexennium. To determine the final assessment of this variable, a weighted mean was calculated using Equation 2. The data for the four universities corresponding to the three factors $(X_1, X_2, X_3)$ were assigned weights $(w_1, w_2, w_3)$ based on their importance.

$$\overline{X} = \frac{w_1 X_1 + w_2 X_2 + w_3 X_3}{w_1 + w_2 + w_3}$$
(2)

Table 5 shows the three factors of the corresponding variable (triennium, quinquennium, and sexennium) with related weights.

The weights assigned to the components of scientific production—triennium (0.20), quinquennium (0.30), and sexennium (0.50)—were determined based on their relative importance in academic evaluation. The sexennium, as the most comprehensive measure, was given the highest weight due to its focus on high-impact publications over a longer period.

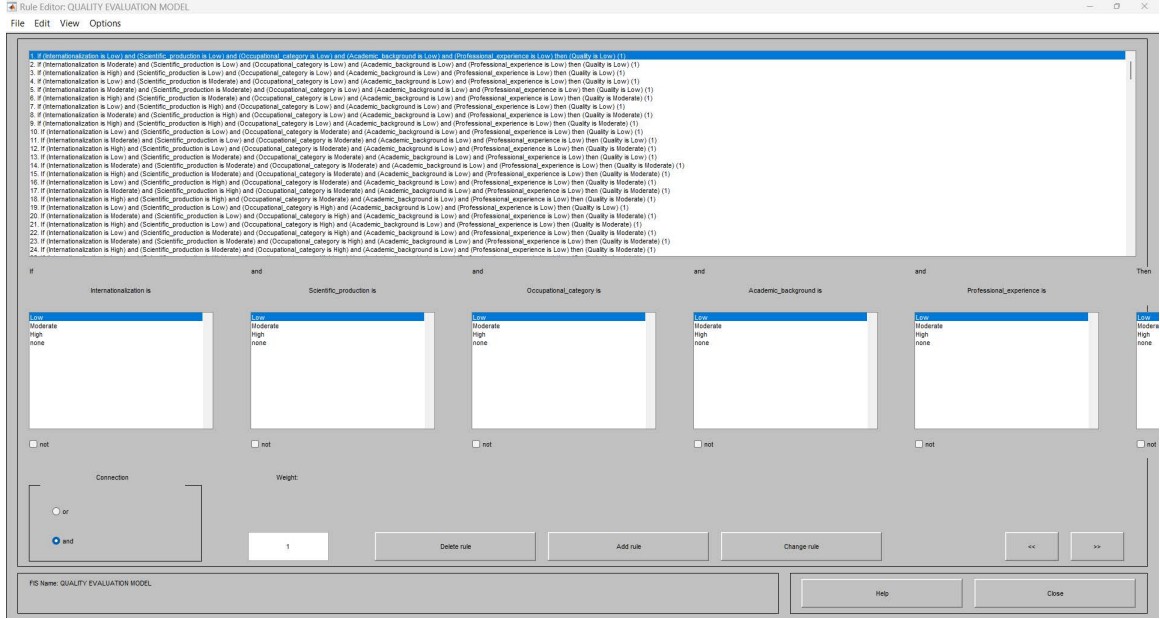

**Fig 4. Configuration of the if-then rules in the Rule Editor.** Source: MATLAB Fuzzy Logic Toolbox Software (MATLAB 9.14 R2023a).

Table 5. Weights applied to the factors of the scientific production variable.

| Factors of the Academic Production Variable | Weight |
|---|---|
| Triennium ($w_1$) | 0.20 |
| Quinquennium ($w_2$) | 0.30 |
| Sexennium ($w_3$) | 0.50 |

Source: own elaboration.

The quinquennium and triennium received proportionally smaller weights because they covered shorter periods and fewer contributions. This weighting approach is consistent with established academic evaluation practices and ensures that the sum of all weights equals one, maintaining consistency within the model.

- <u>Academic background.</u> This variable has two components: people with a doctorate and those without. It represents the percentage of faculty who have obtained a doctorate.

- <u>Occupational category.</u> This variable is made up of various factors depending on the university, classified into two groups: academic staff who hold a permanent position and those who do not; see Table 1. Related job contract types within each category are detailed below (original names in Spanish retained for better understanding):

  - <u>Non-tenure workers</u>: 'ayudante doctor' and 'contratado doctor'.

  - <u>Tenure staff</u>: 'titular de universidad' and 'catedrático de universidad'.

It should be noted that the variables above measure the percentage of faculty members who hold long-term positions, whether they are civil servants (either 'titular de universidad' or 'catedrático de universidad') or not ('contratado doctor').

1. Professional experience. This variable comprises two parts: teaching staff who combine their teaching duties with another professional activity outside the academic realm and those who do not. The ratio of professional research and teaching staff to the total number of academics was calculated.

Each of the institutions studied was assigned a single final score for each input variable on a three-value scale. The final scores were then entered into the FIS for evaluation. As shown in Table 6, the final scores of the input variables for each university have been summarized for a better understanding of the research findings.

## Application of the FIS developed as a model for assessing higher education quality

The processes involved in designing the FIS to assess the quality of university education were detailed in previous sections. The following describes its operation and implementation. Fig 5 illustrates the structure of the developed FIS, which consists of three main components: a) *fuzzification*, b) inference, and b) *defuzzification* [44].

a. *Fuzzification*: As the initial stage in which crisp input values are transformed into fuzzy values based on the linguistic terms established for each input [46].

b. Fuzzy inference process: Following *fuzzification*, the fuzzy input values are processed using the predefined if-then rules to produce fuzzy output values [47].

c. *Defuzzification*: The final stage involves converting the fuzzy output values back into crisp values for practical applications or decision making [48]. Different methods such as centroid, mean of maximum (MOM) and smallest of maximum (SOM) can be used for *defuzzification* to obtain a single crisp output value [49].

In summary, the FIS takes uncertain fuzzy input and uses a set of predefined rules to produce fuzzy output, which are then converted back into crisp values through *defuzzification* for decision-making purposes.

**Table 6. Final crisp evaluations of the university input variables included in the study.**

| Variables | URJC | UC3M | UCM | UAM |
|---|---|---|---|---|
| Internationalization | 0.09 | 0.21 | 0.06 | 0.12 |
| Scientific Production | 0.97 | 0.86 | 3.00 | 1.55 |
| Occupational category | 0.81 | 1.20 | 1.04 | 1.20 |
| Academic background | 2.34 | 2.40 | 2.70 | 2.46 |
| Professional experience | 1.11 | 1.35 | 1.08 | 0.81 |

Source: own elaboration.

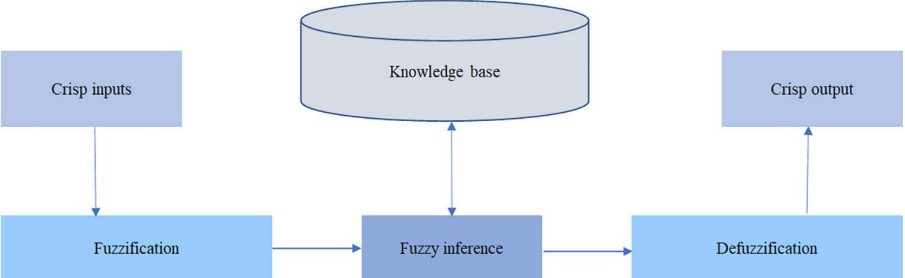

**Fuzzy Inference System (FIS)**

**Fig 5. The structure of the FIS is presented as a model to evaluate the quality of higher education.** Source: Moreno-Cabezali [45].

For the FIS of this investigation, the centroid *defuzzification* method was chosen. It is often considered the optimal choice for any membership function due to several advantages [50]:

-Facilitates a smooth transition within the fuzzy output region.

-Its output calculation is relatively straightforward.

It applies to the geometries of both fuzzy and single-ton output sets. This technique calculates the centroid of the fuzzy set along the x-axis, representing the point at which the fuzzy set would achieve equilibrium. The centroid is computed using the formula specified in Equation 3, where μ ($x_i$) represents the membership function within the universe of discourse [49].

$$Centroid = \frac{\sum_i \mu\,(x_i)\,x_i}{\sum_i \mu\,(x_i)}$$

(3)

Within the MATLAB Fuzzy Logic Toolbox, using the Rule Viewer is required to observe the fuzzy inference process. As illustrated in Fig 6, users using this model can adjust the values of input variables (internationalization, scientific production, occupational category, academic background, and professional experience) and observe the resulting output of each fuzzy rule, the fuzzy set of aggregate output, and the *defuzzified* output value. Consequently, the *defuzzified* output value indicates the level of quality of higher education at each university.

Due to the similarity of the results obtained with the Rule Viewer, it was decided to directly utilize the *readfis*, input, output, and evaluation functions in the MATLAB command window (S1 File) to obtain more accurate results for classifying the universities according to their quality level. This process was carried out as follows:

i) Firstly, five arrays named "A", "B", "C", "D" and "E" were created, respectively, for each of the five input variables. This allowed the data in Table 5 to be directly incorporated into the software.

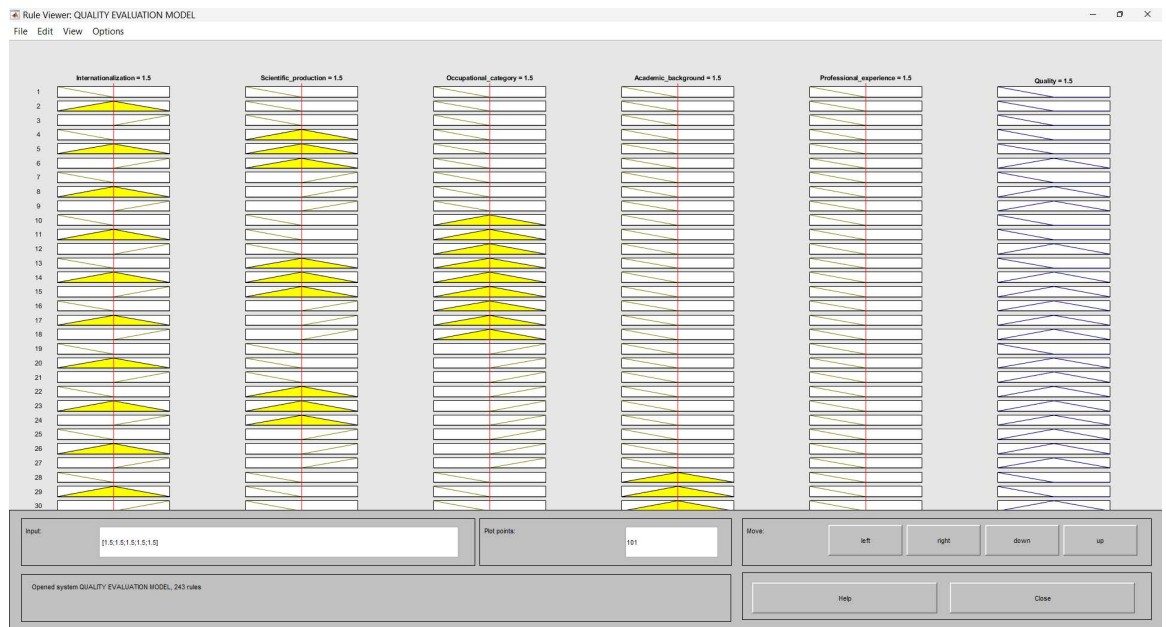

**Fig 6. Rule viewer.** Source: MATLAB Fuzzy Logic Toolbox Software MATLAB Fuzzy Logic Toolbox Software (MATLAB 9.14 R2023a).

ii) Secondly, the developed FIS was uploaded as a model to assess teaching quality at four public universities in Madrid. The *readfis* function, which reads a previously developed FIS file, was used for this purpose.

iii) Thirdly, the *input* function was used to specify the input to the FIS.

iv) Finally, the *evalfis* function was used to evaluate the FIS and obtain the corresponding result for a given set of inputs, which in this case is the level of university education.

## Results and discussion

Universities must be aware of the quality of their educational programs to identify strengths, capabilities, and areas for improvement. In addition, it is a priority to pursue policies that promote academic careers and maintain the quality of higher education, thus improving the future employability of graduates. In this context, this study introduces an innovative tool as a model to assess the quality of higher education from the perspective of teaching staff. As indicated above, a partial analysis was performed in four institutions in Madrid. URJC, UC3M, UAM, and UCM. Thus, Table 7 displays the results produced from the implementation of the FIS based on the evaluations of the input variable.

According to Table 6, the four public universities in Madrid analyzed exhibit very similar values for the quality of university teaching from the faculty perspective. The university with the highest quality level is UCM, with a score of 1.50 on a three-value scale. Consequently, UCM falls into the quality categories 'Low', 'Moderate', and 'High.' The degree of membership for the fuzzy sets 'low' and 'high' is 0, while for the fuzzy set 'Moderate', it is 1. To calculate the degree of membership, Equation 1 was used. Thus, it can be inferred that UCM has a moderate quality, since its degree of membership in the category 'Moderate' is higher than in the categories 'High' and 'Low'.

The next four universities in the ranking are UCM, UC3M, UAM and URJC, with quality grades of 1.4631, 1.4617, and 1.3970 on a three-value scale, respectively. These universities are classified into the 'low' and 'moderate' groups. Specifically, their membership degrees for the "Low" category are 0.025, 0.026, and 0.069, respectively. For the category 'Moderate', their membership degrees are 0.975, 0.974, and 0.931, respectively. Therefore, it can be concluded that these three universities possess a more moderate quality than a low one, since their degree in the category of 'Moderate' membership is higher than those in the category of 'Low'. Based on the degree of membership the four universities have in the "Moderate" group higher; it can be concluded that they have a quality of university education of medium level. Therefore, this analysis can be visually displayed as shown in Fig 7. Specifically, this figure illustrates the degrees of membership of UAM in the 'high' and 'moderate' fuzzy sets of the output variable. Similarly, this representation applies to the remaining universities. Furthermore, based on this analysis, a summary table is shown in Table 8.

The data presented in Table 8 reveal notable disparities among categories within each university, although comparable results among the categories of the four universities show homogeneity. To perform a thorough analysis, these differences can be categorized according to each variable, as described below.

**Table 7. The quality level of higher education in Madrid universities.**

| University | Internationalization | Scientific Production | Occupational category | Academic background | Professional experience | Quality | Classification |
|---|---|---|---|---|---|---|---|
| UCM | 0.06 | 3 | 1.04 | 2.70 | 1.08 | 1.50 | 1 |
| UC3M | 0.21 | 0.86 | 1.20 | 2.40 | 1.35 | 1.4631 | 2 |
| UAM | 0.12 | 1.55 | 1.20 | 2.46 | 0.81 | 1.4617 | 3 |
| URJC | 0.09 | 0.97 | 0.81 | 2.34 | 1.11 | 1.3970 | 4 |

Source: own elaboration.

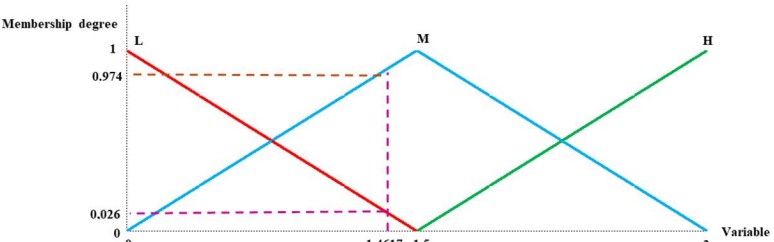

**Fig 7. Membership degrees of the UAM university in the 'high' and 'Moderate' fuzzy sets of the output variable.** Source: own elaboration based on information from MATLAB Fuzzy Logic Toolbox software (MATLAB 9.14 R2023a).

**Table 8. Summary of variables.**

| Variables | Factors | UCM | UC3M | UAM | URJC |
|---|---|---|---|---|---|
| Internationalization | National | 97.53% | 93.30% | 96.25% | 97.07% |
| | Abroad | 2.47% | 6.70% | 3.75% | 2.93% |
| Scientific Production | Triennium | 32,054 | 7,353 | 13,166 | 8,704 |
| | Quinquennium | 13,127 | 3,353 | 5,991 | 4,037 |
| | Sexennium | 8,764 | 2,336 | 4,279 | 2,308 |
| Occupational category | Non-Tenure | 62.67% | 49.36% | 56.30% | 57.47% |
| | Tenure | 34.98% | 40.20% | 39.67% | 27.18% |
| Academic background | Ph.D. | 90.00% | 80.35% | 82.27% | 78.10% |
| | Other training | 10.00% | 19.65% | 17.73% | 21.90% |
| Professional experience | Combine teaching with another job | 35.82% | 45.45% | 27.02% | 36.99% |
| | Does not combine teaching with another job | 64.18% | 54.55% | 72.98% | 63.01% |

Source: own elaboration.

i) <u>Internationalization</u>. In this section, there are no significant differences. UC3M is the university with the lowest number of foreign faculty staff, but, as can be seen in the data, the four universities maintain similar figures, which shows the low level of foreign faculty in these universities.

ii) <u>Scientific production</u>. Here, it should be noted that percentage results can be meaningless whose merits are evaluated through two variables: years and high-impact scientific publications. Thus, an academic can be eligible to receive up to six sexenniums, six quinquenniums, and six trienniums. Thus, it has been considered more enlightening to present these data as the total of triennium, quinquennium, and sexennium.

Concerning the analysis of this variable, significant disparities emerge among the three factors. Across the four universities, there is a notable prevalence of trienniums, followed by quinquenniums, and lastly sexenniums in terms of differentiated productivity. Although faculty staff can apply for and obtain, for example, a sexennium without having applied for either a triennium or a quinquennium, it makes sense that the order is this because if someone has, for example, five high-quality publications in six years (sexennium), he or she will also have four high-quality publications in five years (quinquennium) and three high-quality publications in three years (triennium).

UCM stands out in scientific production compared to the other three universities, in any of the three options of the scientific output: sexennium, quinquennium, and triennium. The second position is UAM, and the other two universities, UC3M and URJC, are almost on par with the last position. These are shown below (see Figs 8–10). However, when comparing the occupational categories with research variables and factors considered, there are differences between the four

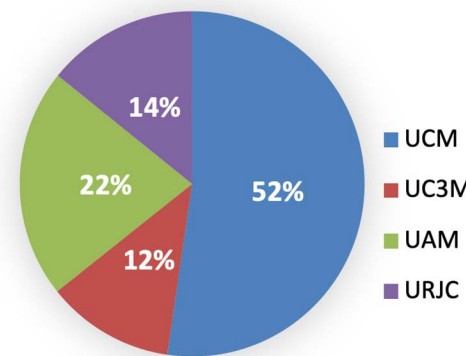

**Fig 8. Triennium per university.** Source: own elaboration.

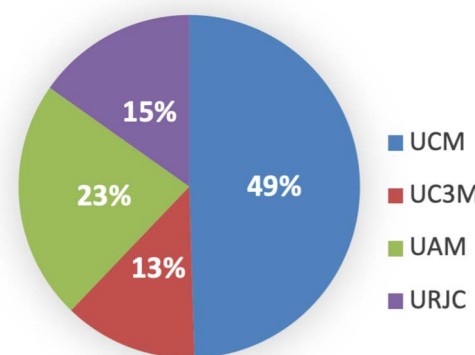

**Fig 9. Quinquennium per university.** Source: own elaboration.

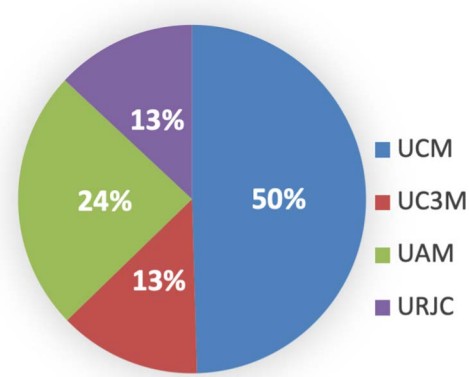

**Fig 10. Sexennium per university.** Source: own elaboration.

universities, it has been observed that the position of 'professor catedrático' is similar in UCM, UC3M, and UAM, while in URJC it is practically half that of any of the other three.

Regarding the 'professor titular', all are above 20%, again the URJC being the one with the lowest percentage of this type of faculty staff. Therefore, in terms of tenure, the URJC is the one with the lowest percentage, compared to the rest of the universities. Regarding non-tenure positions, in general, UCM stands out, with the highest variation of 13 percentage points with UC3M. Within each category, the contrast is striking for the category of 'professor contratado doctor' in UC3M, which practically has none, compared to UAM, with a difference of approximately 17 percentage points. Apparently, the data that stand out most is that of UC3M, with a very marked contrast between the 'profesor asociado' and the other two figures included as 'non-tenure professional contractado doctor' and 'professional ayudante doctor'.

iii) <u>Occupational category</u>. In this category, five faculty staff positions have been considered (see Table 9). Not all records held by each of the universities represented in this study. The reasons for this decision are given in Section 3. Therefore, the total percentage of tenure and non-tenure will not be 100%. The five positions in the analysis have been classified as tenured and non-tenured based on the stability of the position, since 'profesor catedrático' and 'profesor titular' in Spain are positions that once obtained are held for life.

iv) <u>Academic background</u>. This category shows similar data among universities, where all of them have more than 75% of faculty with Ph.D.

v) <u>Professional experience</u>. This category is relevant due to the number of university educators who balance their teaching duties with other professional commitments. These individuals provide students with a practical understanding of the real-world scenarios that they may encounter after graduation. This category mainly includes instructors within higher education faculty who have professional backgrounds. Examining Table 7 reveals a substantial percentage of faculty members that fall into this category at different universities. In most cases, this percentage is close to or even exceeds one third of the faculty. At UC3M, it represents almost half of the faculty, 45.45%.

## Research limitations and future directions

Among the limitations encountered in conducting this analysis, several have been identified: a) The first of the limitations has been identified in the analysis of the starting information needed to carry out the research. Although it is true that all universities are public and therefore are governed by the corresponding legislation, cloisters with different denominations have been found, so previous work has been done to identify the equivalences between them; b) the second limitation has been found in the language since there are different positions in Spain compared to other countries, so the information has been checked with colleagues from faculties in the U.S.; c) the third and last limitation has been identified with the universities under study. This study has focused on quality measurement in public universities in Madrid (Spain). The data was extracted from UniversiDATA, a platform that extracts and unifies data from the four universities in the analysis

**Table 9. Occupational categories.**

| Occupational category (Spain) | Occupational category (U.S) | FACTORS | UCM | UC3M | UAM | URJC |
|---|---|---|---|---|---|---|
| Profesor catedrático | Professor/full professor | Tenure | 11.77% | 12.56% | 14.63% | 6.59% |
| Profesor titular | Associate professor | Tenure | 23.21% | 27.64% | 25.05% | 20.59% |
| Profesor contratado doctor | Associate Professor | Non-tenure | 13.45% | 0.45% | 18.19% | 13.14% |
| Profesor ayudante doctor | Assistant Professor | Non-tenure | 13.39% | 3.46% | 11.29% | 7.35% |
| Profesor asociado | Professional professor | Non-tenure | 35.82% | 45.45% | 27.02% | 36.99% |

Source: own elaboration.

to make them comparable with each other. Four of the six public universities in the capital were included in this analysis. This was due to the difficulty of finding and comparing data from the other two universities.

As for the future research agenda, once the analysis has been carried out and the gaps to be filled have been identified, it is proposed: i) to carry out an analysis using a Mamdani FIS, where the quality of the universities is investigated from a different perspective, that of the student body, focusing on variables such as the level of students who carry out exchanges with other universities, the internship programs, or the level of a second language required at the time of completing the university degree; ii) Considering that this study has already made the equivalences between the faculty in Spain and the US, another future line of research could be to replicate this same analysis with the same variables in any group of US universities with similar characteristics between them.

The main value of this study lies in its design, which uses fuzzy logic to provide a flexible and effective way to handle the complexity of assessing higher education quality, particularly when dealing with subjective or imprecise data. This design allows the methodology to be extended and applied to the study of any other university, adapting to diverse contexts and systems. However, the main limitation of the article is the difference in competence between countries and universities. When determining the variables, it will be necessary to consider both general and specific characteristics. The internationalization of university teaching staff is a variable strongly affected by socio-economic factors, which means that in future studies it should be considered in a dichotomous context, as in other countries it may result from positive trends. Therefore, this study is seen as the beginning of a research line whose next step is to extend the proposed methodology to the study or to generalize it to other universities and countries.

## Conclusions

With the European panorama of higher education, the concept of quality was placed at the center of all attention. Quality evaluation in European Higher Education Institutions (HEI) became a key issue among academics and policymakers [2]. Quality assurance in higher education is a generic term open to many interpretations. Despite the rapid proliferation of international ranking systems in recent years, criticisms of cross-border university comparisons have emerged. The existing literature has not agreed on the best methodology for university quality assessment. Through the design of a Mamdani FIS implemented in MATLAB Fuzzy Logic Toolbox, this paper presents a novel methodology for quality assessment of universities, concretely of Spanish universities in Madrid, based on the characteristics of the human capital of HEIs. This methodology has not been used before to analyze the quality of higher education, so it is a great contribution to the literature.

Based on the existing literature, five variables were selected to construct the proposed metric: internationalization, scientific production, occupational category, academic background, and professional experience. In addition, this methodology has been applied and the quality of four universities in Madrid has been evaluated. UAM, UCM, UC3M, and URJC. The methodology has placed UCM, UC3M, UAM, and URJC in that order. Although the difference in quality levels between them is very small, it is important to bear in mind that the UCM is the oldest university in the capital city of Madrid, and the URJC is the newest of all the public universities in Madrid. Keeping this in mind will help us better interpret the data obtained in this analysis. This methodology not only serves to establish university rankings but also has important practical implications for universities since it can be proposed as a point of reference for universities to know their strengths and weaknesses to maintain their strengths and focus their attention on reinforcing their weaknesses so that they can achieve higher levels of quality. By using fuzzy logic, the methodology balances complexity and practicality, making it adaptable to diverse contexts and university systems. All should improve the internationalization variable, concerning the other categories, and based on the results obtained. The UCM maintains a very high scientific production as a strength but should focus on improving the occupational category. With UC3M, it is necessary to improve scientific production, and it stands out in the professional experience category, because it has a high number of professional professors, according to the assessment made in this research regarding this variable, it is a positive point to provide students with this theoretical-practical perspective. The strength of the UAM is found in the academic background, while the professional experience stands out as

its weak point. Regarding the URJC, the newest of the universities analyzed, it stands out in the academic background and should improve the occupation category by expanding the tenured staff, which in turn would be an opportunity to increase scientific production. The higher the position among faculty staff, the higher the number of scientific publications. It is striking that all universities studied share a weakness in the internationalization category.

Another of the practical implications identified in the article is its high replicability, since, in addition to allowing replicating the analysis in universities in other countries with similar elements and systems, it provides the possibility of replicating the analysis by comparing the quality of university education between Spanish territories. This flexibility allows the methodology to serve as the foundation for future studies expanding to universities in different countries and contexts. This brings us to the last practical implication of this study, since the analysis conducted can help policymakers design better practices to improve the careers of university professors and, consequently, the quality of higher education and the future employability of graduates.

Concerning theoretical implications. Up to now, there has been a correlation between the four variables mentioned above and university quality. This observed relationship had already been consolidated by the literature, but individually, that of each variable separately from university quality. Therefore, one of the main contributions of this research is the performance of a joint analysis of variables and their relationship with university quality.

In addition, there are several studies on the quality of higher education institutions from the perspective of the student and not from the perspective of the professors, so another contribution to this article is the adoption of this new approach, both in the perspective considered to evaluate quality and, in the methodology, used to do so.

The FIS designed and applied in the present study has proven to be a suitable tool to evaluate university quality from the perspective of its human capital, so the objective envisaged in this research has been achieved. Its ability to address the subjectivity and complexity of teaching perceptions provides a solid foundation for informed decision-making and the continuous improvement of HEIs.

## Supporting information

**S1 File. MATLAB Command Window.** Source: MATLAB Fuzzy Logic Toolbox Software (MATLAB 9.14 R2023a). (XLSX)

## Author contributions

**Conceptualization:** Cristina Carrasco-Garrido, Antonio Martínez Raya.

**Data curation:** Cristina Carrasco-Garrido, Belen Maria Moreno-Cabezali.

**Formal analysis:** Cristina Carrasco-Garrido, Belen Maria Moreno-Cabezali.

**Investigation:** Cristina Carrasco-Garrido, Belen Maria Moreno-Cabezali.

**Resources:** Cristina Carrasco-Garrido, Belen Maria Moreno-Cabezali.

**Methodology:** Cristina Carrasco-Garrido, Belen Maria Moreno-Cabezali.

**Project administration:** Antonio Martínez Raya.

**Software:** Belen Maria Moreno-Cabezali.

**Supervision:** Antonio Martínez Raya.

**Validation:** Antonio Martínez Raya.

**Visualization:** Belen Maria Moreno-Cabezali.

**Writing – original draft:** Cristina Carrasco-Garrido, Belen Maria Moreno-Cabezali, Antonio Martínez Raya.

**Writing – review & editing:** Cristina Carrasco-Garrido, Antonio Martínez Raya.

Funding acquisition: Cristina Carrasco-Garrido.

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
