## [Decision Letter · Decision Letter 0]

10 Nov 2024

PONE-D-24-33158New Perspectives on University Quality Assessment: A Mamdani Fuzzy Inference System ApproachPLOS ONE

Dear Dr. Martínez Raya,

Thank you for submitting your manuscript to PLOS ONE. After careful consideration, we feel that it has merit but does not fully meet PLOS ONE’s publication criteria as it currently stands. Therefore, we invite you to submit a revised version of the manuscript that addresses the points raised during the review process.

We look forward to receiving your revised manuscript.

Kind regards,

Cigdem Kadaifci, Assoc. Prof.

Academic Editor

PLOS ONE

**Journal Requirements:**

Laura Molero-González is supported by the grant PID2021-127836NB-I00 from the Ministerio español de Ciencia e Innovación and Fondo Europeo de Desarrollo Regional (FEDER).

Reviewers' comments:

Reviewer's Responses to Questions

**Comments to the Author**

1. Is the manuscript technically sound, and do the data support the conclusions?

Reviewer #1: Yes

Reviewer #2: Yes

2. Has the statistical analysis been performed appropriately and rigorously? 

Reviewer #1: Yes

Reviewer #2: Yes

3. Have the authors made all data underlying the findings in their manuscript fully available?

Reviewer #1: Yes

Reviewer #2: No

4. Is the manuscript presented in an intelligible fashion and written in standard English?

Reviewer #1: Yes

Reviewer #2: Yes

5. Review Comments to the Author

**Reviewer #1:**  This paper discusses the quality of higher education in general with the specific case of the city-state of Madrid, Spain. Even so, the case seems to represent all European countries based on the considerations submitted to quality assurance institutions, of course with the rules attached to each other.

1. However, it is undeniable that each country or university has its own competence, and perhaps determining the measurement variables must consider the general and specific characteristics for it, which are not explained from the perspective of the interests of each university, then a methodology is proposed to assess the quality of the university.

2. Perhaps, not all causal patterns in language are appropriate to the variables considered, if not directly related to the pattern is the decision that causes the value of the results.

3. The results may not be directly quantitative, but the quantity is a reflection of a methodology that the measurement in the study has been successful. Perhaps, the author considers expanding the proposed methodology, by reinterpreting the measurement variables.

4. Although this study is a view of what is being studied, the synchronization of the existing problems between the potential objectives so that the conclusions answer the problems raised by the author.

Overall this paper is good. Thank you for the opportunity to read it.

**Reviewer #2: ** Dear Author(s),

Assessing the quality of higher education is indeed a difficult task because there is a counterargument and a counterapproach approach for every argument, method, and approach to assessment.

The value of the article is determined by the design of the study, so this approach can be scaled—to apply such a research design to any University. However, expecting a positive return from complex methods is not always necessary.

Despite the fact that the choice of five variables is based on previous research, the internationalization of the university teaching staff may have various reasons. In some countries or regions, this process is driven by negative socio-economic factors, while in others, it may be the result of positive trends. Therefore, this variable can be considered in a dichotomous context when extending the study's design to other countries or regions. – Authors can specify this aspect in the Limits section.

Given the subjective nature of assessing the quality of higher education, more than five variables should be considered. However, the choice of these variables is probably dictated by the available data set. There are various proxy metrics for assessing the quality of education that have a scientific basis, but a significant number of studies in the scientific community are devoted to assessing the quality of education based on the knowledge, skills, and abilities of both students and teachers. That is why proxy metrics do not always show an objective assessment.

It has been suggested to fix two disadvantages:

Page 18: In formula No. 1, there is a 'si' in the fourth rule – this is left over from the translation from Spanish to English = 'if'.

Page 20: "Scientific production. This variable includes three components: triennium, quinquennium, and sexennium. To determine the final assessment of this variable, a weighted mean was calculated using Equation (2). The data for the four universities corresponding to the three factors ( 1, 2, 3) were assigned weights ( 1, 2, 3) based on their importance."

How are weights assigned? ( 1, 2, 3) three factors (X1, x2, x3). The paper presents Table 5 with weights, but it is not clear how the weights were obtained. What is clear is that the sum must be equal to one.

Nevertheless, the article has its own scientific significance and contribution to developing higher education assessment.

6. PLOS authors have the option to publish the peer review history of their article (what does this mean? ). If published, this will include your full peer review and any attached files.

**Do you want your identity to be public for this peer review?** For information about this choice, including consent withdrawal, please see our Privacy Policy .

Reviewer #1: **Yes: ** Mahyuddin Khairuddin Matyuso Nasution

Reviewer #2: No

---

## [Author Response · Author response to Decision Letter 1]

7 Feb 2025

The manuscript has been properly enriched with significant improvements according to the valuable comments provided from peer-review process, both those written by Editor and Reviewers. We hope that is to the liking of all them. Thank you for your attention.

---

## [Decision Letter · Decision Letter 1]

28 Feb 2025

New Perspectives on University Quality Assessment: A Mamdani Fuzzy Inference System Approach

PONE-D-24-33158R1

Dear Dr. Martínez Raya,

We’re pleased to inform you that your manuscript has been judged scientifically suitable for publication and will be formally accepted for publication once it meets all outstanding technical requirements.

Kind regards,

Cigdem Kadaifci, Assoc. Prof.

Academic Editor

PLOS ONE

Additional Editor Comments (optional):

I have reviewed the revisions the second reviewer requested to ensure they have been addressed. Based on my assessment, the authors have sufficiently addressed the reviewers' concerns. My final decision is to accept the manuscript.

Reviewers' comments:

Reviewer's Responses to Questions

**Comments to the Author**

1. If the authors have adequately addressed your comments raised in a previous round of review and you feel that this manuscript is now acceptable for publication, you may indicate that here to bypass the “Comments to the Author” section, enter your conflict of interest statement in the “Confidential to Editor” section, and submit your "Accept" recommendation.

Reviewer #2: All comments have been addressed

2. Is the manuscript technically sound, and do the data support the conclusions?

Reviewer #2: Yes

3. Has the statistical analysis been performed appropriately and rigorously? 

Reviewer #2: Yes

4. Have the authors made all data underlying the findings in their manuscript fully available?

Reviewer #2: No

5. Is the manuscript presented in an intelligible fashion and written in standard English?

Reviewer #2: Yes

6. Review Comments to the Author

Reviewer #2: Dear Authors,

The disadvantages have been eliminated.

Additional information about data:

According to the PLOS Data Policy, authors are required to make all data underlying the findings of their manuscript fully available without restriction, except in rare cases (refer to the Data Availability Statement in the manuscript PDF file for details). The data should be provided either as part of the manuscript and its supporting information or deposited in a public repository.

Currently, the dataset is available from the corresponding author upon reasonable request. However, to ensure compliance with PLOS guidelines and facilitate transparency and reproducibility, we recommend making the data publicly accessible through a trusted repository such as Mendeley Data or Figshare.

Please create a file or files containing the dataset extracted from UniversiDATA, analyzed in this article, and provide the repository link in the article. If using Mendeley Data, upload your dataset there and include the corresponding link in the Data Availability Statement. Also kindly add the direct link in the "Analysis of data and definition of linguistic variables" paragraph.

Best regards,

Reviewer

7. PLOS authors have the option to publish the peer review history of their article (what does this mean? ). If published, this will include your full peer review and any attached files.

**Do you want your identity to be public for this peer review?** For information about this choice, including consent withdrawal, please see our Privacy Policy .

Reviewer #2: No

---

## [Editor Report · Acceptance letter]

PONE-D-24-33158R1

PLOS ONE

Dear Dr. Martínez Raya,

I'm pleased to inform you that your manuscript has been deemed suitable for publication in PLOS ONE. Congratulations! Your manuscript is now being handed over to our production team.

Kind regards,

on behalf of

Dr. Cigdem Kadaifci

Academic Editor

PLOS ONE
